# Low Health Literacy Is Associated with Poorer Physical and Mental Health-Related Quality of Life in Dialysed Patients

**DOI:** 10.3390/ijerph192013265

**Published:** 2022-10-14

**Authors:** Ivana Skoumalova, Andrea Madarasova Geckova, Jaroslav Rosenberger, Maria Majernikova, Peter Kolarcik, Daniel Klein, Andrea F. de Winter, Jitse P. van Dijk, Sijmen A. Reijneveld

**Affiliations:** 1Department of Health Psychology and Research Methodology, Faculty of Medicine, P.J. Safarik University, Trieda SNP 1, 040 01 Kosice, Slovakia; 2Graduate School Kosice Institute for Society and Health, Faculty of Medicine, P.J. Safarik University, Trieda SNP 1, 040 01 Kosice, Slovakia; 3Department of Community & Occupational Medicine, University Medical Center Groningen, University of Groningen, Antonius Deusinglaan 1, 9713 AV Groningen, The Netherlands; 4Institute of Applied Psychology, Faculty of Social and Economic Sciences, Comenius University Bratislava, Mlynské luhy 4, 821 05 Bratislava, Slovakia; 5Olomouc University Social Health Institute, Palacky University, Univerzitni 22, 771 11 Olomouc, Czech Republic; 6FMC-Dialysis Services Slovakia, Trieda SNP 1, 040 01 Kosice, Slovakia; 7II. Internal Clinic Faculty of Medicine, P.J. Safarik University, Trieda SNP 1, 040 01 Kosice, Slovakia; 8Institute of Mathematics, Faculty of Science, P. J. Safarik University, Jesenna 5, 040 01 Kosice, Slovakia

**Keywords:** health-related quality of life, mental component score, physical component score, health literacy, dialysed patients

## Abstract

Health-related quality of life (HRQoL) is an important health indicator in chronic diseases like kidney diseases. Health literacy (HL) may strongly affect HRQoL, but evidence is scarce. Therefore, we assessed the associations of HL with HRQoL in dialysed patients. We performed a cross-sectional study in 20 dialysis clinics across Slovakia (n = 542 patients, mean age = 63.6 years, males = 60.7%). We assessed the association of categorised HL (low, moderate, high) with the SF36 physical component score (PCS) and mental component score (MCS) using generalised linear models adjusted for age, gender, education, and comorbidity (Charlson Comorbidity Index, CCI). We found significant associations of HL with PCS and MCS in dialysed patients, adjusted for age, gender, education, and CCI. Low-HL patients had a lower PCS (B = −3.27, 95%-confidence interval, CI: −5.76/−0.79) and MCS (B = −6.05, 95%-CI: −8.82/−3.29) than high-HL patients. Moderate-HL patients had a lower MCS (B = −4.26, 95%-CI: −6.83/−1.69) than high-HL patients. HL is associated with physical and mental HRQoL; this indicates that dialysed patients with lower HL deserve specific attention and tailored care to have their HRQoL increased.

## 1. Introduction

In recent decades, chronic kidney disease (CKD) has become one of the main public health problems along with cardiovascular disease, obesity, HIV, hypertension, and malaria [1]. CKD is an irreversible process of renal impairment and affects 10–15% of the world population [2,3,4]. According to the Global Burden of Disease study (GBD, 2019), CKD was the 18th leading cause of death globally in 2019 for all ages and even the 8th leading cause of death for the age groups of 50 years and over [5]. The burden of disease due to CKD measured as disability-adjusted life years has increased considerably during the last 30 years, with CKD ranking among the top 10 burdens. Great effort is taken to combat the multiple environmental, genetic, behavioural, and socioeconomic determinants and causes of CKD [1,6,7]. The treatment of CKD, mainly in advanced stages requires a substantial amount of public finances [8]. In high income countries, treating patients with stage 5 CKD generates approximately 2–3% of all healthcare expenditures [4]. CKD thus remains a major challenge for health policies [6]. Along with the aforementioned, CKD considerably affects patients’ health-related quality of life (HRQoL), due to the loss of the functions of the kidneys and related complications, such as high blood pressure, proteinuria, anaemia, bone weakness, fluid retention, hyperkalaemia, and others [9]. These complications and disease-related symptoms affect HRQoL in all aspects, including functional health, psychological well-being, and social life [10]. 

HRQoL is an important outcome of patient-centred care as well as an important predictor of clinical outcomes in chronic diseases, like kidney diseases [11]. HRQoL is defined as the effect of illness and its treatment on a patient’s life as it is perceived by the patient [12] and it is linked to the bio-psycho-social model of health and illness [13]. HRQoL encompasses multiple aspects, such as the perceived burden of the disease, the effect of the symptoms of the disease on the performance of daily activities, the quality of social interaction with relevant others, perceived social support from relatives and healthcare professionals, and emotional well-being. Dialysed patients suffer from physical symptoms such as pain, fatigue, nausea, and frailty and from the impact of these symptoms on their daily performance at work, at home, and on the ability to be engaged in social life activities [14,15]. Moreover, the treatment itself imposes a heavy burden due to the strict diet and fluid intake limitations, and frequent visits to dialysis which challenge self-management because of their complexity and high demand [16]. Living with such health condition as CKD affects patients emotionally, due to perceived burden of the disease, loss or significant reduction of various activities and possibilities in life, and also due to changes in personal and family life [17]. On top, many clinical factors affect HRQoL in CKD patients, such as the modality of treatment [18,19,20], comorbidity [21,22], stage of the disease [22], and the number and severity of symptoms [23]. Moreover, HRQoL is associated with psychological factors, such as depression and anxiety [24], and health literacy (HL).

Low HL is common in CKD patients and may negatively affect HRQoL, in particular in the advanced stages of CKD. Advanced stages of CKD, such as stage 5 CKD place great burden on patients regarding treatment, self-management, and interactions with health care providers, and low HL may hamper patients´ ability to adapt to complex situations, to collaborate with healthcare providers, and to manage the disease and the treatment effectively. Sørensen et al. [25] (p. 3) defined HL as a “person’s knowledge, motivation, and competences to access, understand, appraise and apply health information in order to make judgments and take decisions in everyday life concerning health care, disease prevention, and health promotion to maintain or improve quality of life during the life course”. Low HL is linked to various adverse health outcomes in dialysed patients, such as the lower hazard of referral for transplant evaluation, higher rates of hospitalisations, missed dialysis sessions, higher mortality, and onset of CKD during the life course [26,27,28,29], but the relation has not consistently been confirmed regarding HRQoL in stage 5 CKD patients. Green et al. [30] using a tool that measured functional HL did not find a significant association between HL and HRQoL in dialysed patients. Other studies that used a multidimensional measurement tool to assess HL (i.e., the Health Literacy Questionnaire) showed that HL was associated with poorer HRQoL in dialysed patients [31] and in the stages 3–5 CKD patients with lower HL [32]. Dodson et al. [31] also found a significantly worse HRQoL in dialysed patients with lower HL compared to patients with higher HL. Stømer et al. [32] found that patients in stages 3–5 CKD (14% of the sample were dialysed patients) with low HL had a significantly worse HRQoL than patients with high HL. Patients with moderate HL scored significantly better in mental HRQoL than patients with low HL, but significantly worse in physical HRQoL than patients with high HL. All of the aforementioned studies compared groups of patients according to the level of their HL (two or three HL groups), as measured using different tools for HL and for HRQoL. The evidence on the association of HL and HRQoL is inconsistent regarding other patient groups (e.g., cardiovascular disease, psoriasis, and frequent users of health services with at least one chronic condition), i.e., some studies show significant association and some non-significant association [33,34,35].

A multidimensional assessment of HL, which exceeds the functional dimension of HL (i.e., reading skills, basic comprehension and basic numeracy skills) seems to be more accurate for detecting the key areas connected with the quality of life in dialysed patients, but evidence on this is very limited. Dodson et al. [31] addressed this with a relatively small study sample of 76 haemodialysed patients and Stømer et al. [32] in a study comprising only 26 dialysed patients. Therefore, the aim of our study is to assess whether multidimensionally measured HL is associated with mental and physical HRQoL in a large cohort of dialysed patients. We expect that patients with lower HL will report worse mental and physical HRQoL than patients with higher HL, also if gender, age, and comorbidity are taken into account.

## 2. Materials and Methods

### 2.1. Sample and Procedure

We collected our data from January to November 2018 within a network of 20 dialysis clinics in Slovakia. These dialysis clinics belong to a private dialysis network—Fresenius Medical Care-dialysis services in Slovakia—and cover about 20% of the total Slovak dialysis population. The costs of the care as provided is routinely included in and financed by the Slovak health insurance system. We included patients over 18 years with a diagnosis of stage 5 CKD and who had been undergoing dialysis for at least 3 months. We excluded patients who were not able to fill in the questionnaires (due to a psychiatric diagnosis or an inability to read the Slovak language) and those who had an acute severe intercurrent illness, similar to other studies performed in such patient populations [36]. 

Data were obtained using questionnaires and by extracting from medical records. Patients were approached during their visit to their dialysis centre and asked to participate in the study. After signing an informed consent, they filled in online questionnaires using tablets. Data were recorded with full confidentiality assured by means of a personal identification code. The study was approved by the Ethics Committee of the Faculty of Medicine of P. J. Safarik University (15N/2017) and the Ethics Committee of the Fresenius Medical Care-dialysis services.

### 2.2. Measures

Health-related quality of life was measured using the Rand 36-item health survey (SF-36), which is a part of the Kidney Disease Quality of Life-Short Form (KDQoL-SF, [37]). Cronbach’s Alphas of the questionnaire´s scales ranged from 0.54 to 0.94. We used the physical component score (PCS), and the mental component score (MCS). We further denote these as physical and mental HRQoL, respectively. The higher the score of the MCS and PCS the better the quality of life. The mean scores of these components were used for further analyses, similar to other studies [38].

Health Literacy was measured using the Health Literacy Questionnaire (HLQ, Slovak version, [39]) which is a nine-dimensional tool for measuring health literacy competencies related to appraising, understanding and using health information for further decisions about health [40]. Cronbach´s Alphas in our sample ranged from 0.77 to 0.90. We categorised the HLQ using hierarchical cluster analysis, which resulted in 3 HL clusters, groups of patients with a similar construction of health literacy, respectively. The detailed procedure of handling with this variable is described elsewhere [41]. These 3 HL groups were used for further analyses (low, moderate, high HL group). 

Background data regarded age, gender, education (lower: elementary and apprenticeship vs. higher, i.e., secondary and university) and comorbidity. The first two were measured using the questionnaire. For comorbidity, we used the age-adjusted Charlson Comorbidity Index (CCI; [42,43]) based on data obtained from the medical records of the patients. The score of this measurement tool ranges from 0–33, with higher scores indicating more comorbidities. 

### 2.3. Statistical Analyses

First, we assessed the sociodemographic and clinical characteristics (CCI) as well as the MCS and PCS, overall and by HL group (Table 1). Second, we assessed the association of categorised HL (low, moderate, high) with the PCS and MCS of dialysed patients using Generalised Linear Models (GLM), crude (Table 2, Model 1) and adjusted for age (continuous level), gender, education (dichotomized), and CCI (continuous level) (Model 2). For each outcome (PCS and MCS), we report regression coefficients (B) with 95%-confidence intervals (CI). A *p*-value of <0.05 was assumed for statistical significance. Statistical analyses were performed with SPSS v.23.0 for Windows, IBM Corp., Armonk, NY, USA [44].

## 3. Results

### 3.1. Baseline Characteristics

We included 567 patients on maintenance dialysis. The final sample consisted of 542 patients after excluding those where data on HL were partly or completely missing (n = 25). The mean age of the respondents was 63.6 years (range: 21–91 years) and more than half of the respondents were males (60.7%); 32% had low HL, 54% had moderate HL and 14% had high HL. The mean CCI score was 6.5.

### 3.2. Associations of Health Literacy with Health-Related Quality of Life

We found significant associations of HL with HRQoL in dialysed patients after adjustment for age, gender, education, and comorbidity. Patients with low HL were more likely to have a lower PCS (B −3.28, 95%-CI: −5.76/−0.79, *p* < 0.01) and a lower MCS (B −6.05, 95%-CI: −8.82/−3.29, *p* < 0.001) in comparison to patients with high HL. Patients with moderate HL were more likely to have a lower MCS (B −4.26, 95%-CI: −6.83/−1.69, *p* < 0.001) in comparison to high-HL patients. We did not find any significant association regarding moderate HL and PCS in comparison with the high HL group.

## 4. Discussion

We found that lower HL is associated with poorer HRQoL in dialysed patients. Patients with low HL were more likely to have a lower physical HRQoL than patients with high HL. We also found that patients with low and moderate HL were more likely to have a worse mental HRQoL. Findings of this study contribute to current knowledge regarding the association between HL and HRQoL.

We found that lower HL is associated with worse physical and mental HRQoL in dialysed patients. Our findings are consistent with the findings of the study by Dodson et al. [31], who found that patients with lower HL had significantly lower physical and mental HRQoL. However, we measured HL in three categories, while Dodson et al. [31] used only two categories (lower and higher HL). Our use of three categories showed that patients with moderate HL had a worse mental HRQoL than patients with higher HL, but that these two categories did not differ in physical HRQoL. The mental HRQoL is thus even poorer in patients who have a relatively good health literacy, i.e., a moderate level. This suggests that patients with moderate HL are still not protected against the mental effects of the disease and its treatment even though they know better than patients with low HL how to obtain and understand health information and communicate about health issues with healthcare providers. From a clinical perspective, this suggests a need to address emotional and psychological needs in patients with low as well as moderate HL as these patients are at risk of poor mental HRQoL. Moreover, it suggests that the capacity of understanding in patients with moderate HL may be overestimated by clinicians. This issue is of particular importance as poor mental HRQoL has been shown to be associated with a higher risk of death and higher rates of hospitalisation in end-stage renal disease patients [45].

Our findings are also partly consistent with those of Stømer et al. [32] in their study on HL and HRQoL in stage 3–5 CKD patients, 14% of whom were dialysed patients. The difference is that we did not find a significant association between moderate HL and physical HRQoL. These findings suggest that in dialysed patients low HL is associated with lower HRQoL and more strongly with its mental than physical aspects, whereas in slightly less advanced stages of CKD the associations of HL with physical HRQoL may be somewhat stronger. An explanation of these differences may be that the longer the disease and the treatment last, the greater their negative effect is on mental HRQoL in patients with lower levels of HL (low and moderate) than in patients with high HL. Patients with lower levels of HL also have lower self-efficacy [46], and thus self-efficacy may be a factor contributing to poorer mental HRQoL [47,48]. This association requires further study.

Our finding of lower HL being associated with worse HRQoL is inconsistent with that of Green et al. [30], who did not report a significant association of HL with HRQoL. An explanation may be that Green et al. measured HL with a one-dimensional tool focusing on the ability to read and pronounce medical terms (REALM). This suggests that more complex skills related to HL (e.g., an engagement with healthcare providers, effective navigating of the healthcare system, ability to find appropriate health information) are more important in relation to HRQoL in dialysed patients than a basic comprehension of written health information and reading and numeracy skills, which are referred to as a functional HL. One-dimensional questionnaires can overestimate or underestimate patient’s HL, as they focus on only one aspect of HL. In contrast, we assessed HL using a multidimensional tool, which measures various domains of HL beyond the reading, and numeracy skills related to health information, similar to Dodson et al. [31] and Stømer et al. [32]. An explanation of our findings then may be that poorer skills and capacities in HL in dialysed patients contribute to a worse HRQoL probably due to having insufficient information and a poorer understanding of health information and treatment recommendations. This can lead to being less active in managing one’s health and being less able to effectively communicate on health concerns or difficulties in adhering to treatment with the healthcare providers. However, there is not much evidence on which HL domains, in particular, are associated with worse HRQoL in CKD or in other patient groups, and our study also did not assess this issue. Existing research shows that HLQ 9 [32], HLQ 4, 6–9 [49], and HLQ 1, 2, 5, 6–9 [35] were associated with the quality of life in studied groups of patients. This may be interpreted as that different diseases or different stages of disease put different challenges to using HL skills.

The major strength of our study is the use of a large nationally representative sample of dialysed patients, covering approximately 20% of dialysed patients in Slovakia within 20 dialysis centres, with a relatively high response rate (70%) that enable our findings to be generalised for the population of dialysed patients. Another strength is that we used a multidimensional tool to measure HL (HLQ), which allows us to capture this concept more comprehensively, beyond reading and writing skills, and the SF-36 to measure HRQoL, which is a frequently used measurement tool to measure the quality of life in patients with chronic diseases. Both instruments show good validity and reliability. 

Our study also has some limitations, the most important being that it was cross-sectional which does not allow us to draw conclusions about causality. Another limitation is that patients with the worst level of HL and the worst level of HRQoL probably were not included in the study sample, which may have led to some underestimation of the associations between HL and HRQoL. Moreover, data on HL and HRQoL were self-reported, potentially leading to some additional measurement error and thus less accurate estimates. 

Our finding that lower HL is associated with worse HRQoL implies that CKD patients with lower HL deserve specific attention when receiving haemodialysis, either by improving the patients’ HL skills or by increasing the ability of healthcare providers to identify and recognise patients with lower HL skills and offering them greater support and attention.

We found the HRQoL of dialysed patients with lower health literacy to be worse for both its physical and mental components in comparison with patients with high HL. This shows a need to pay more attention to patients with lower HL, as they are at greater risk of suffering from worse HRQoL. This may be carried out by early and careful identification of the level of HL using multidimensional tools to assess HL in patients with CKD [39,40], and by improving healthcare providers´ competencies in recognising low HL, such as using red flags in the interview (e.g., patient is inactive in asking questions, doesn´t understand oral or written health information) [50,51]. A further idea could be to help patients with lower HL in understanding health information, to verify their understanding repeatedly, and to support them in expressing their doubts about treatment or other related problems regarding self-management, taking medication, diet, and coping with the consequences of the disease [52]. Research has shown that this can be effectively carried out using visual strategies, and by supporting rather individual discussions than group discussion of needs and barriers [53]. Finally, a promising way to tackle the impact of low HL on HRQoL is to focus on intermediate factors between HL and HRQoL such as smoking [54]. Moreover, our findings imply that mental HRQoL is poor even when patients have a moderate level of HL skills. For clinical care this finding implies a need to address the psychological needs of both patients with low and moderate HL, such as stress, anxiety, depression, uncertainty, and feelings of being a burden for healthcare professionals and for their relatives. This may be carried out by providing patients with psychological care as a part of routine renal care.

Future research should focus on the association between multidimensional HL and HRQoL in earlier stages of the disease as well as on its association with changes in HRQoL during the course of the disease to identify if lower HL might be a factor that contributes to the decrease in the HRQoL. Such a longitudinal approach may also shed more light on the question of whether contextual factors play a mitigating role in the relation of HL with HRQoL. It would be relevant to examine the role of social support, the level of HL skills in patients’ caregivers, and the degree of health literacy responsiveness of healthcare providers and healthcare organisations, such as the use of plain language or offering more time and support to patients with low HL. Moreover, more research is needed on factors that may mediate the relationship between HL and HRQol, such as adherence, physical activity, alcohol, and substance abuse, etc. [54].

## 5. Conclusions

Lower HL is associated with poorer HRQoL in dialysed patients. Improving or at least maintaining the level of HRQoL of these patients requires paying more attention to their HL needs and limitations in relation to their treatment and self-management. 

## Figures and Tables

**Table 1 ijerph-19-13265-t001:** Characteristics of the sample: gender, age, education, CCI, PCS, and MCS, overall and by 3 HL groups, frequencies or means (n = 542, patients from 20 dialysis clinics in Slovakia interviewed in 2018).

			Health Literacy	
Characteristics	Overall (n = 542)	Low (n = 172; 31.7%)	Moderate (n = 293; 54.1%)	High (n = 77; 14.2%)
Sociodemographic				
Male gender (n, %)	329 (60.7)	105 (61.0)	181 (61.8)	43 (55.8)
Age (mean ± SD)	63.6 ± 14.12	65.3 ± 14.3	62.9 ± 13.8	62.5 ± 14.7
Lower education (n, %)	266 (49.1)	92 (17.0)	139 (25.6)	35 (6.5)
Clinical				
Charlson Comorbidity Index ^1^ (mean ± SD)	6.5 ± 2.9	6.8 ± 2.8	6.4 ± 2.9	6.1 ± 3.0
Health-Related Quality of Life ^2^				
Physical Component Score (mean ± SD)	34.1 ± 9.7	31.7 ± 9.6	35.1 ± 9.5	36.7 ± 10.0
Mental Component Score (mean ± SD)	45.2 ± 10.3	43.3 ± 9.6	45.2 ± 10.4	49.4 ± 10.8

^1^ Scores range from 2 to 16; missing cases = 6. ^2^ missing cases for PCS and MCS = 4.

**Table 2 ijerph-19-13265-t002:** Association of health literacy with PCS and MCS, crude effect (Model 1) and adjusted for age, gender, education, and CCI (Model 2) (GLM, n = 542).

	Model 1 (Crude)		Model 2 (Adjusted)	
	PCS B (95% CI)	MCS B (95% CI)	PCS B (95% CI)	MCS B (95% CI)
Health literacy				
Low HL	−3.96 (−6.54/−1.38) **	−6.05 (−8.78/−3.31) ***	−3.28 (−5.76/−0.79) **	−6.05 (−8.82/−3.29) ***
Moderate HL	−0.55 (−2.96/1.85)	−4.15 (−6.70/−1.59) ***	−0.41 (−2.72/1.90)	−4.26 (−6.83/−1.69) ***
High HL	Ref.	Ref.	Ref.	Ref.

** *p* < 0.01, *** *p* < 0.001.

## Data Availability

The dataset used and analysed for this study is available from the corresponding author on reasonable request.

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
