# Peer review of "Low Health Literacy Is Associated with Poorer Physical and Mental Health-Related Quality of Life in Dialysed Patients"

_ijerph, 2022, doi:10.3390/ijerph192013265_

Round 1

Reviewer 1 Report

Thanks for the opportunity to review the manuscript. Overall, the article reads well. Yet, it could benefit from additional information and clarification. Below I provided feedback for authors to consider.

Introduction

Line 59-61. The meaning of the sentence is not clear. Clarification is needed.  

Line 76. What are other factors? You may provide some examples.  

Line 82-85. The sentence structure may need to be revised for clarification.

Line 88. What are other factors? You may provide some examples.  

Methods

Measures: Please report score ranges and reliability of all measures.   

Analyses: Please add rationale of controlling for only gender, age, and comorbidity while not including other socioeconomic factors (e.g., income, education, insurance, etc.) or any differences in sites/clinics (e.g., rural vs. urban). If you have this information, you may include in your analyses.

Results

Please provide the range of participants’ age.

Discussion

Line 197-201. The sentence seems not clear. Also, the authors could elaborate the clinical implications by adding some factors associated with mental HRQoL.

Line 238-240. The authors should be cautious with this statement as they did not provide sampling information to ensure that study participants are a representative sample. Also, based on the article, it appears that survey questionnaires were completed by voluntary participants which may bias the findings.   

Reviewer 2 Report

This is a well written paper that uses well established tools for measuring quality of life and different aspects of health literacy. However, it would be interesting to learn something new about this association, and move us forward in our understanding of the simple association between health literacy and quality of health.

My main concern is the missing measure of education, education and health literacy as strongly associated, and as we know education is associated with different aspects of health’ therefore it is important to know if health literacy adds on to this association or is the association dependent on level of education and not health literacy.

One issue that could move us forward could be how to improve health literacy or treat low health literacy differently but except of one paragraph at the end of the discussion this is not contend with.

Specific issues

Page 2 row 89-90:  The health literacy measure used by Green et al (29) is a compeatly different measure of health literacy, it measures functional health literacy and no what is measured in this paper. This should be explicitly written.

Page 4 table 1: why not add P values to shoe the difference between the levels of health literacy?

Page 5 row 222-223: “…and are therefore inaccurate and unsuitable for practical use” I think this sentence is miss leading, the measures measure different things’ and this does not mean the functional health literacy measured by REALM is inaccurate. It is just something else.

Page 6- The authors suggest further research looking at other variables such as social support, I could not agree more and I think this is missing in this paper.

Also the paragraph on page 6 row 257-273 is very important and further research should go there and not perform more studies showing the same results similar to previous studies.

Round 2

Reviewer 2 Report

no additional comments